# Different methods of fear reduction are supported by distinct cortical substrates

**Belinda PP Lay[1], Audrey A Pitaru[1], Nathan Boulianne[1], Guillem R Esber[2], Mihaela D Iordanova[1]***

[1]Center for Studies in Behavioural Neurobiology, Department of Psychology, Concordia University, Montreal, Canada; [2]Department of Psychology, Brooklyn College of the City University of New York, Brooklyn, United States

**Abstract** Understanding how learned fear can be reduced is at the heart of treatments for anxiety disorders. Tremendous progress has been made in this regard through extinction training in which the aversive outcome is omitted. However, current progress almost entirely rests on this single paradigm, resulting in a very specialized knowledgebase at the behavioural and neural level of analysis. Here, we used a dual-paradigm approach to show that different methods that lead to reduction in learned fear in rats are dissociated in the cortex. We report that the infralimbic cortex has a very specific role in fear reduction that depends on the omission of aversive events but not on overexpectation. The orbitofrontal cortex, a structure generally overlooked in fear, is critical for downregulating fear when novel predictions about upcoming aversive events are generated, such as when fear is inflated or overexpected, but less so when an expected aversive event is omitted.

## Introduction

Extinction learning has captivated behavioural and neural science for more than a century. It has done so because it allows for the reduction of behaviours that were once adaptive but are no longer so, and gives the therapist a handle to combat others that were never adaptive in the first place. The most-widely used method for suppressing unwanted behaviour relies on the *omission* of the event that drives this behaviour; that is extinction. In the context of fear learning, extinction involves the dramatic reduction in fear-related behaviours typically observed after presenting a previously established signal for an aversive event (i.e., a tone paired with shock; tone →shock) in the absence of that event (tone presented alone; tone → nothing). Given its simplicity and effectiveness in the treatment of anxiety disorders (*Schiller et al., 2010*; *Milad and Quirk, 2012*; *Craske et al., 2018*; *Ebrahimi et al., 2020*; *Geller et al., 2019*; *Hammoud et al., 2020*), extinction has received significant attention in a quest to understand its underlying behavioural and neural mechanisms (e.g., *An et al., 2017*; *Morgan et al., 1993*; *Quirk et al., 2000*; *Milad and Quirk, 2002*; *Leung and Westbrook, 2008*; *Leung and Westbrook, 2010*; *Likhtik et al., 2008*; *Herry et al., 2008*; *Monfils et al., 2009 Johansen et al., 2011*).

Critically, although much progress has been made, this quest has focused on a single method for reducing learned fear: that involving outcome omission in the presence of a previously conditioned cue, while another, equally relevant method that drives reduction in conditioned behaviour; namely, overexpectation, remains largely unexplored. This single-paradigm approach is restrictive because at best it can oversimplify and at worst even misrepresent the function of brain areas implicated in extinction learning. Here, we move beyond this paradigm-specific approach and embark on an investigation into how the brain learns to reduce learned fear using two behavioural designs: extinction driven by the omission of an expected aversive event (described above) and overexpectation driven by generating novel predictions that surpass the delivered aversive event (described below).

*For correspondence:
mihaela.iordanova@concordia.ca

In overexpectation, reduction in previously established fear responses ensue, strikingly, despite continued delivery of the aversive event. This is possible because separately established signals of a common aversive event (i.e., tone → shock; light→shock) can summate their fear-inducing properties when encountered simultaneously (tone+light), triggering a state of exacerbated fear. The presentation of the same (i.e., unintensified) aversive event in that state (tone+light →shock) engages a self-regulatory mechanism that lessens the exacerbated fear by partially extinguishing the fear elicited by each individual signal (*Rescorla, 1970*; *Rescorla, 2006*; *Rescorla, 2007*). Since signals for threat often co-occur (think of the sight of a microphone and that of a staring crowd for a glossophobic), failure to reduce fear by overexpectation is a likely contributor to the aberrant and persistent fear characterizing anxiety disorders such as panic disorder, social anxiety disorder or post-traumatic stress disorder.

The infralimbic cortex (IL) is considered to be the key brain locus in learning to downregulate fear responses (*An et al., 2017*; *Do-Monte et al., 2015*; *Laurent and Westbrook, 2009*; *Lingawi et al., 2017*; *Milad and Quirk, 2002*; *Milad et al., 2004*; *Meyer and Bucci, 2014*; *Quirk et al., 2000*; *Sierra-Mercado et al., 2006*; *Sierra-Mercado et al., 2011*). Importantly, evidence for this role is exclusively derived from extinction designs. If the IL is critical for learning to reduce fear in general, then it should do so irrespective of the conditions that generate this reduction. It follows that in our dual-paradigm approach, the IL should mediate fear reduction in both extinction and overexpectation. Alternatively, the IL might have a paradigm-specific role in learning that leads to suppression of established behaviour when expected outcomes are omitted. To date, there is no evidence to bear on either alternative. To address this gap in knowledge, we first examined whether the IL is necessary for reduction in fear driven by overexpectation and then by extinction.

In our study of the neurobiology of fear reduction, we also chose to move beyond the traditional fear circuit. Our candidate was the lateral orbitofrontal cortex (lOFC), a structure strongly linked to reward learning (e.g., *Gardner et al., 2019*; *Padoa-Schioppa and Assad, 2006*; *Rich and Wallis, 2016*). We targeted the OFC for three reasons. Firstly, it has dense reciprocal projections with the basolateral amygdala (*Carmichael and Price, 1995*; *Price, 2007*), which has been extensively implicated in fear acquisition and extinction (e.g., *Davis, 2000*; *Courtin et al., 2014*; *Herry et al., 2008*; *Kim and Davis, 1993*; *Laurent et al., 2008*; *Orsini and Maren, 2012*). Secondly, the lOFC has been linked to fear (*Zelinski et al., 2010*; *Asok et al., 2013*; *Trow et al., 2017*; *Chang et al., 2018*; *Ray et al., 2018*; *Zimmermann et al., 2018*) and anxiety (*Etkin and Wager, 2007*; *Milad and Rauch, 2007*). Finally, it supports learning from overexpectation of reward (*Burke et al., 2009*; *Takahashi et al., 2009*). Therefore, we examined the role of the lOFC in reduction of learned fear responses in overexpectation and extinction.

## Results

### Experiment 1: The IL regulates learning from extinction but not overexpectation

Experiment 1 examined the effect of IL inactivation on reducing learned fear driven by overexpectation and extinction. Rats were implanted with cannula targeting the IL bilaterally (*Figure 1A and B*) and trained to associate a tone with shock (tone → shock) and a flashing light with shock (flash →shock, *Figure 1C*). Percent time spent freezing to the cues was taken as a measure of the level of conditioned fear (*Blanchard and Blanchard, 1969*; *Fanselow, 1980*). Fear conditioning was acquired to the tone and the light (*Figure 1D*). A mixed ANOVA revealed no effect of training (tone: $F_{(1, 45)} = 0.18$, $p = 0.68$, 95% CI [-0.55, 0.39]; flash: $F_{(1, 45)} = 0.09$, $p = 0.77$, 95% CI [-0.41, 0.52]), no effect of drug (tone: $F_{(1, 45)} = 0.035$, $p = 0.85$, 95% CI [-0.51, 0.43]; flash: $F_{(1, 45)} = 0.46$, $p = 0.50$, 95% CI [-0.34, 0.59]), and no interaction (tone: $F_{(1, 45)} = 0.84$, $p = 0.36$, 95% CI [-0.30, 0.64]; flash: $F_{(1, 45)} = 0.035$, $p = 0.85$, 95% CI [-0.50, 0.43]). Freezing increased across days and the rate of acquisition was similar across groups as shown by a significant linear trend (tone: $F_{(1, 45)} = 69.77$, $p < 0.001$, $\eta_p^2 = 0.61$, 95% CI [1.20, 1.96]; flash: $F_{(1, 45)} = 166.96$, $p < 0.001$, $\eta_p^2 = 0.79$, 95% CI [1.79, 2.46]), and no linear trend x training x drug interaction (tone: $F_{(1, 45)} = 0.069$, $p = 0.79$, 95% CI [-0.84, 1.04]; flash: $F_{(1, 45)} = 0.041$, $p = 0.84$, 95% CI [-0.75, 0.88]).

Following robust fear conditioning to the individual cues, the tone and the light were presented in compound and paired with the same single shock as that delivered during initial fear acquisition in

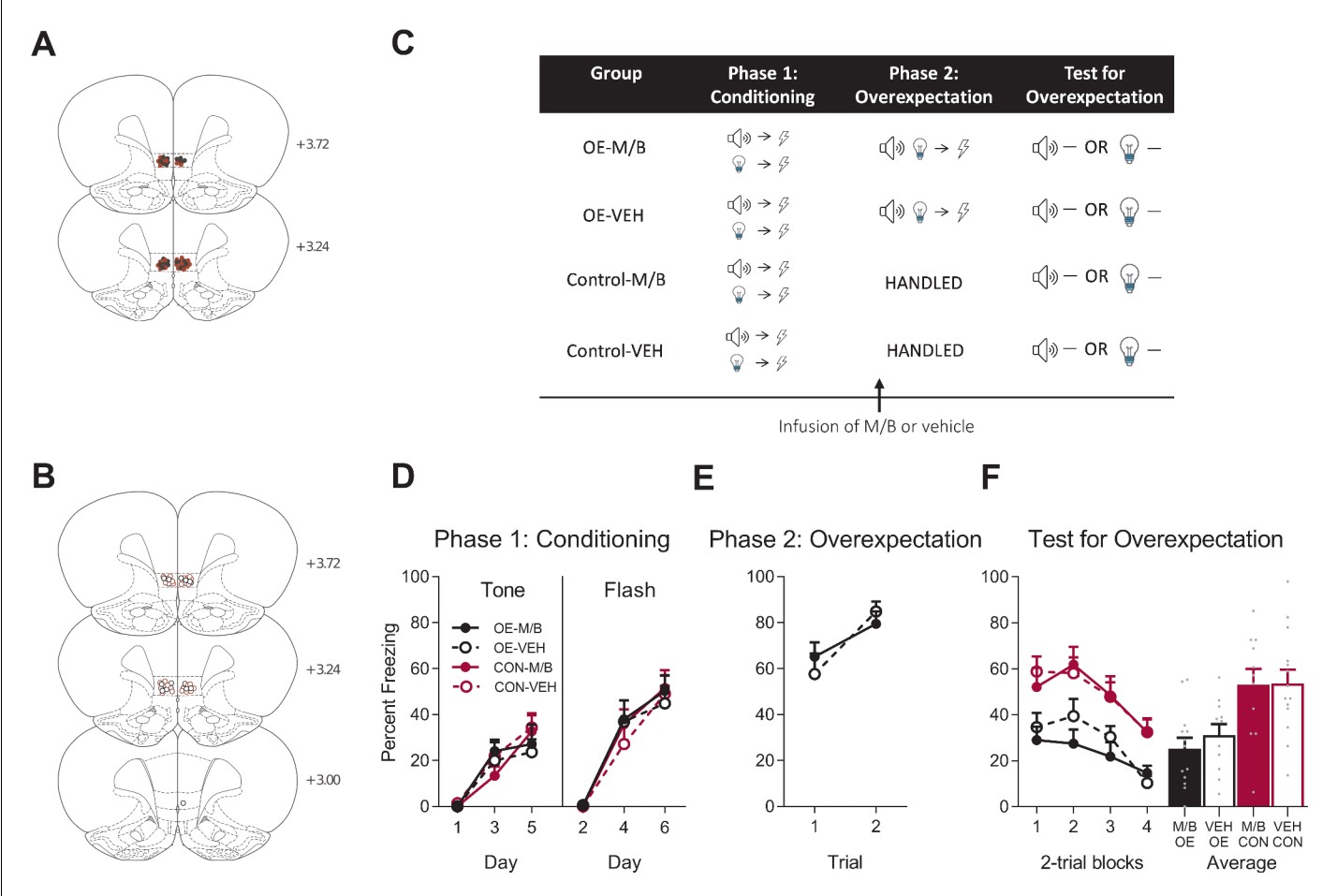

**Figure 1.** The IL is not necessary for overexpectation. Location of cannula placements for (A) drug- and (B) vehicle-infused rats in the IL cortex in the overexpectation experiment as verified based on the atlas of *Paxinos and Watson, 1997*. The symbols represent the most ventral point of the cannula track for each rat and distances are indicated in mm from bregma. (C) Behavioural design for Overexpectation. Behavioural data are represented as mean + SEM percent levels of freezing during the cue period for D) Conditioning, (E) Overexpectation and, (F) Test for Overexpectation of the target stimulus. Overexpectation-M/B (OE-M/B, filled black), n = 13; overexpectation-vehicle (OE-VEH, open black), n = 11; control-M/B (CON-M/B, filled burgundy), n = 11; control-vehicle (CON-VEH open burgundy), n = 14.

The online version of this article includes the following source data for figure 1:

**Source data 1.** The IL is not necessary for overexpectation.

order to generate the overexpectation condition (*Figure 1C*). Prior to overexpectation training the rats received either infusions of muscimol and baclofen (0.1 mM muscimol-1 mM baclofen, M/B) or vehicle into the IL. This was done in order to inactive the IL during overexpectation learning (M/B) and compare its effect to an identical group that received overexpectation training but in the presence of a functional IL (vehicle). Rats in the control conditions did not receive overexpectation training but received identical infusions of the drug or the vehicle (*Figure 1C*). The overexpectation-M/B and overexpectation-vehicle groups exhibited similar levels of responding to the compound stimulus during this phase (*Figure 1E*). A mixed ANOVA revealed no effect of group ($F_{(1, 22)}$=0.022, p=0.88, 95% CI [−0.64, 0.74]), an increase in responding across trials shown by a significant linear trend ($F_{(1, 22)}$=15.10, p=0.001, $\eta_p^2$ = 0.41, 95% CI [0.44, 1.44]) and no linear trend x group interaction ($F_{(1, 22)}$=1.45, p=0.24, 95% CI [−1.59, 0.42]).

During Test, the target cue (tone or light, counterbalanced) was presented alone in the absence of shock (*Figure 1C*). Rats trained in overexpectation with a functional or inactivated IL showed lower level of fear to the target cue compared to the controls (*Figure 1F*). A mixed ANOVA revealed a main effect of training ($F_{(1, 45)}$=19.21, p<0.001, 95% CI [−1.48,−0.41], d = 1.30), no main effect of

drug ($F_{(1, 45)}$=0.33, p=0.57, 95% CI [−0.66, 0.41]), and no training x drug interaction ($F_{(1, 45)}$=0.21, p=0.65, 95% CI [−0.64, 0.44]). There was a significant linear trend across trials ($F_{(1, 45)}$=42.63, p<0.001, $\eta_p^2$ = 0.49, 95% CI [−1.01,–0.53]), confirming a decline in responding across Test but there was no linear trend x training x drug interaction ($F_{(1, 45)}$=0.24, p=0.63, 95% CI [−0.47, 0.70]), confirming a similar decline in responding across all groups. These results indicate that infusion of M/B in the IL cortex prior to compound presentations during overexpectation training had no effect on retrieval of the overexpectation memory when tested drug-free.

Following the null effect of IL inactivation on overexpectation, we wanted to confirm that our inactivation parameters and current placements would replicate the well-established effect of IL inactivation on the reduction of learned fear using extinction in the same rats (*Figure 2C*). The allocation of rats to groups was counterbalanced based on their prior experience in the overexpectation study (see Materials and methods) and their placements in this part of the experiment are depicted as per their new group assignment (*Figure 2A and B*). All rats were conditioned to fear a novel cue (steady light or white-noise, counterbalanced). Fear was acquired to the auditory and visual cues (*Figure 2D*). A mixed ANOVA revealed no main effect of training ($F_{(1, 33)}$=0.19, p=0.67, 95% CI

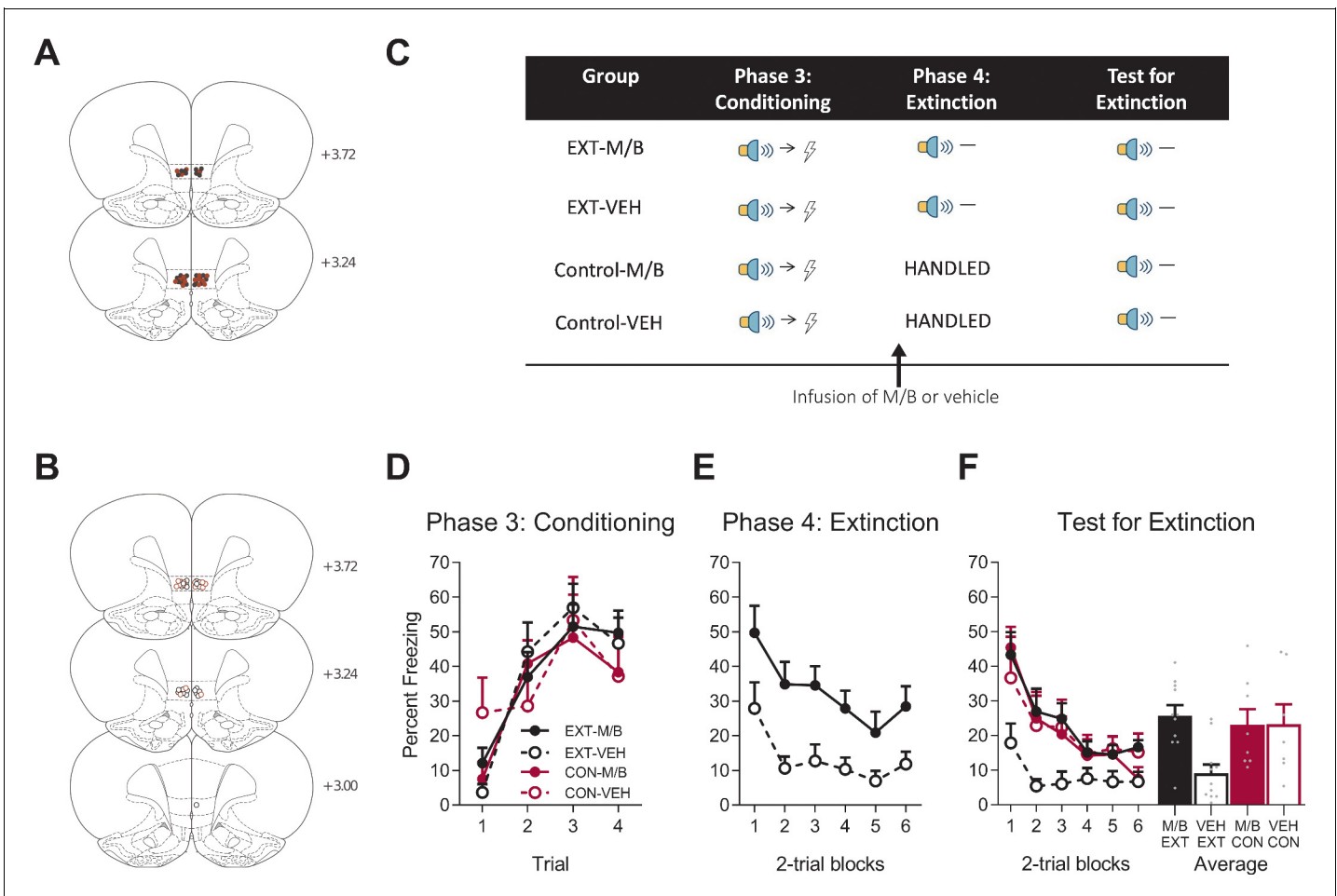

**Figure 2.** The IL is necessary for extinction. Using the same animals, the location of cannula placements reallocated for (A) Drug- and (B) Vehicle-infused rats in the IL extinction experiment as verified based on the atlas of *Paxinos and Watson, 1997*. The symbols represent the most ventral point of the cannula track for each rat and distances are indicated in millimetres from bregma. (C) Behavioural design for Extinction. Behavioural data are represented as mean + SEM percent levels of freezing during the cue period for (D) Conditioning, (E) Extinction, and (F) Test for Extinction of the target stimulus. Extinction-M/B (filled black), n = 11; extinction-vehicle (open black), n = 11; control-M/B (filled burgundy), n = 8; control-vehicle (open burgundy), n = 7.

The online version of this article includes the following source data for figure 2:

**Source data 1.** The IL is necessary for extinction.

[−0.52, 0.74]), no main effect of drug ($F_{(1, 33)}$=0.06, p=0.81, 95% CI [−0.69, 0.57]), and no training x drug interaction ($F_{(1, 33)}$=0.038, p=0.85, 95% CI [−0.58, 0.68]). A significant linear trend ($F_{(1, 33)}$=44.82, p<0.001, $\eta_p^2$ = 0.58, 95% CI [0.76, 1.42]) and no linear trend x training x drug interaction ($F_{(1, 33)}$=0.84, p=0.37, 95% CI [−1.12, 0.52]) indicate that fear increased across trials but the rate of increase was similar across groups.

Fear conditioning was followed by extinction training in half of the cohort. Extinction consisted of non-reinforced presentations of the fear conditioned cue (*Figure 2E*). The remaining half of the cohort received no training during this phase. Prior to extinction training, the rats received either M/B or vehicle infusions into the IL. Identical infusions were also given to the control rats that did not receive extinction. Rat infused with M/B in the IL showed higher levels of fear across extinction trials compared to rats infused with the vehicle (*Figure 2E*). A mixed ANOVA revealed a main effect of drug ($F_{(1, 20)}$=12.80, p=0.002, 95% CI [0.40, 1.54], d = 1.53), a significant linear trend ($F_{(1, 20)}$=9.49, p=0.006, $\eta_p^2$ = 0.32, 95% CI [−1.18,–0.23]) but no linear trend x drug interaction ($F_{(1, 20)}$=0.67, p=0.42, 95% CI [−1.33, 0.58]), indicating that the rate of reduction in responding was similar for both extinction groups.

The following day, all rats were then tested for fear to the target cue in the absence of shock. Silencing the IL prior to extinction training disrupted subsequent retrieval of the extinction memory (*Figure 2F*). A mixed ANOVA revealed no main effect of training ($F_{(1, 33)}$=2.23, p=0.15, 95% CI [−0.80, 0.20]), a main effect of drug ($F_{(1, 33)}$=4.52, p=0.041, 95% CI [−0.08, 0.92], d = 0.89), and a significant training x drug interaction ($F_{(1, 33)}$=4.72, p=0.037, $\eta_p^2$ = 0.13, 95% CI [−0.07, 0.93]). Post-hoc tests with Bonferroni adjustments revealed a significant difference within the extinction conditions: rats infused with M/B during extinction froze significantly more during presentations of the cue on Test compared to rats infused with vehicle ($F_{(1, 33)}$=11.43, p=0.002, 95% CI [0.19, 1.52], d = 1.85); there was no effect of drug on the control condition ($F_{(1, 33)}$=0.001, p=0.98, 95% CI [−0.82, 0.80]). There was a significant linear trend across trials ($F_{(1, 33)}$=40.05, p<0.001, $\eta_p^2$ = 0.55, 95% CI [−1.24,–0.64]) but no linear trend x training x drug interaction ($F_{(1, 33)}$=0.07, p=0.79, 95% CI [−0.89, 0.73]). Taken together, these data provide key evidence for the dissociable role of the IL in fear reduction: IL function is necessary for supporting fear reduction driven by extinction but not overexpectation learning.

## Experiment 2: The lOFC regulates learning from overexpectation but not extinction

In the first part of Experiment 2 we examined the role of the lOFC in overexpectation. Rats received fear conditioning of the auditory and visual cues in a manner identical to that described in Experiment 1 (tone→shock, flash→shock, *Figure 3C*). Conditioned fear was acquired to the auditory and visual cues (*Figure 3D*). A mixed ANOVA revealed no effect of training (tone: $F_{(1, 40)}$ = 0.001, p = 0.97, 95% CI [-0.46, 0.48]; flash: $F_{(1, 40)}$ = 0.52, p = 0.48, 95% CI [-0.39, 0.70]), no effect of drug (tone: $F_{(1, 40)}$ = 0.53, p = 0.47, 95% CI [-0.33, 0.60]; flash: $F_{(1, 40)}$ = 0.76, p = 0.39, 95% CI [-0.35, 0.73]), and no training x drug interaction (tone: $F_{(1, 40)}$ < 0.001, p = 0.98, 95% CI [-0.46, 0.47]; flash: $F_{(1, 40)}$ = 0.66, p = 0.42, 95% CI [-0.37, 0.72]). Freezing increased across days and the rate of acquisition was similar across groups as shown by a significant linear trend (tone: $F_{(1, 40)}$ = 213.40, p < 0.001, $\eta_p^2$ = 0.84, 95% CI [2.51, 3.31]; flash: $F_{(1, 40)}$ = 264.90, p < 0.001, $\eta_p^2$ = 0.87, 95% CI [2.73, 3.50]), and no linear trend x training x drug interaction (tone: $F_{(1, 40)}$ = 0.15, p = 0.70, 95% CI [-0.84, 1.15]; flash: $F_{(1, 40)}$ = 0.007, p = 0.93, 95% CI [-0.92, 0.99]).

Following fear conditioning, rats were infused with M/B or vehicle into the lOFC and then trained in overexpectation (tone+light→shock; *Figure 3C*). Rats in the control conditions received identical infusions but no overexpectation training (see Materials and methods). Infusions of M/B in the lOFC had no effect on freezing to the compound cues during the overexpectation phase (*Figure 3E*). A mixed ANOVA revealed no main effect of drug: $F_{(1, 21)}$ = 0.05, p = 0.83, 95% CI [-0.60, 0.75]), no linear trend ($F_{(1, 21)}$ = 1.42, p = 0.25, 95% CI [-0.23, 0.86]) and no linear trend x drug interaction ($F_{(1, 21)}$ = 2.40, p = 0.14, 95% CI [-1.90, 0.28]), confirming that rats in the overexpectation-M/B and overexpectation-vehicle maintained similar responding to the compound stimulus during this phase.

To determine whether the lOFC was important for learning during overexpectation of fear, rats were tested for fear to the target cue (tone or light, counterbalanced between rats) in the absence of shock (*Figure 3C*). Inactivation of the lateral OFC prior to overexpectation training disrupted the

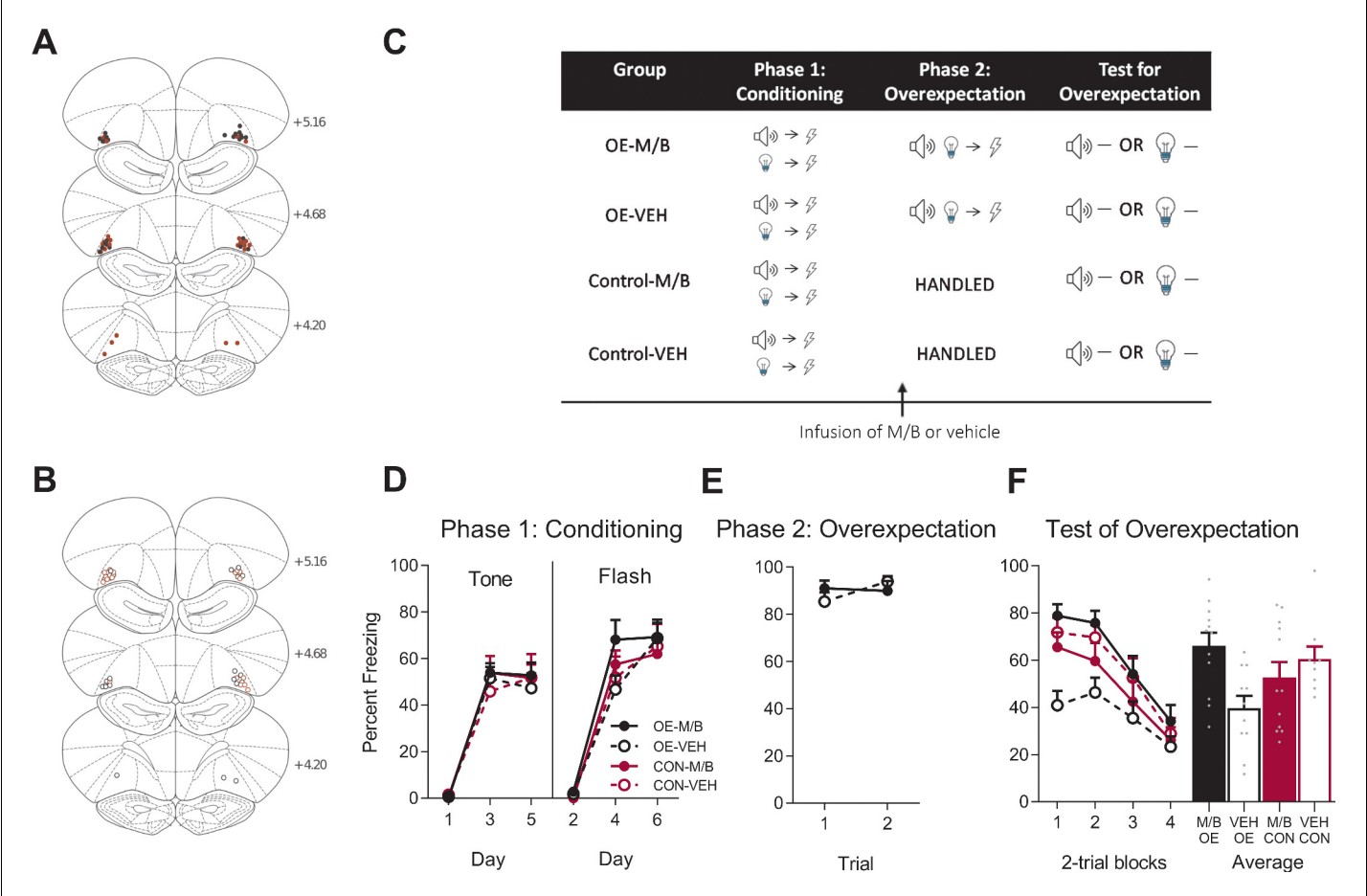

**Figure 3.** The lOFC is necessary for overexpectation. Location of cannula placements for (**A**) drug- and (**B**) vehicle-tinfused rats in the lOFC overexpectation experiment as verified based on the atlas of *Paxinos and Watson, 1997*. The symbols represent the most ventral point of the cannula track for each rat and distances are indicated in millimetres from bregma. (**C**) Behavioural design for Overexpectation. Behavioural data are represented as mean + SEM percent levels of freezing during the cue period for (**D**) Conditioning, (**E**) Overexpectation and, (**F**) Test for Overexpectation of the target stimulus. Overexpectation-M/B (filled black), n = 12; overexpectation-vehicle (open black), n = 11; control-M/B (filled burgundy), n = 12; control-vehicle (open burgundy), n = 9.

The online version of this article includes the following source data for figure 3:

**Source data 1.** The lOFC is necessary for overexpectation.

overexpectation effect on Test (*Figure 3F*). A mixed ANOVA revealed no main effect of training ($F_{(1, 40)}$=0.42, p=0.52, 95% CI [−0.69, 0.40]), no main effect of drug ($F_{(1, 40)}$=2.53, p=0.12, 95% CI [−0.20, 0.89]), but a significant training x drug interaction ($F_{(1, 40)}$=8.73, p=0.005, $\eta_p^2$ = 0.18, 95% CI [0.10, 1.19]). Post-hoc tests with Bonferroni adjustments showed that overexpectation training reduced fear to the target cue in rats with a functional lOFC: Rats infused with vehicle prior to overexpectation training froze significantly less during presentations of the target stimulus on Test compared to the vehicle rats in the control condition ($F_{(1, 40)}$=5.93, p=0.019, 95% CI [−1.63, 0.06], d = 1.26). This overexpectation effect was disrupted in rats trained under an inactivated lOFC: Rats trained in overexpectation and infused with M/B in the lOFC froze significantly more during presentations of the target stimulus at Test compared to the vehicle rats ($F_{(1, 40)}$=10.93, p=0.002, 95% CI [0.21, 1.77], d = 1.44). There was no effect of drug on the controls ($F_{(1, 40)}$=0.88, p=0.35, 95% CI [−1.13, 0.53]). There was a significant linear trend across trials ($F_{(1, 40)}$=84.59, p<0.001, $\eta_p^2$ = 0.68, 95% CI [−1.57,−1.00]), confirming a decline in responding across Test but there was no linear trend x training x drug interaction ($F_{(1, 40)}$=2.84, p=0.10, 95% CI [−1.23, 0.29]), confirming a similar decline in responding

across all groups. These results indicate that, in the absence of the lOFC, rats were not able to reduce conditioned fear responding that is driven by overexpectation training.

Similar to Experiment 1, the same rats took part in an extinction experiment. The allocation of rats to groups was counterbalanced based on their prior experience in the overexpectation study (see Materials and methods) and their placements in this part of the experiment are depicted as per their new group assignment (*Figure 4A and B*). All rats were conditioned to fear a novel cue (steady light or white-noise, counterbalanced, *Figure 4C*). Conditioned fear was acquired to the novel cue (steady light or white-noise, counterbalanced; *Figure 4D*). A mixed ANOVA revealed no main effect of training ($F_{(1, 40)}$=0.003, p=0. 96, 95% CI [−0.57, 0.55]), no main effect of drug ($F_{(1, 40)}$=0.16, p=0.69, 95% CI [−0.65, 0.47]), and no training x drug interaction ($F_{(1, 40)}$=0.015, p=0.90, 95% CI [−0.59, 0.54]). A significant linear trend ($F_{(1, 40)}$=67.52, p<0.001, $\eta_p^2$ = 0.63, 95% CI [0.93, 1.53]) revealed an increase in fear across trials, but this increase did not differ between the groups, as revealed by the absence of a linear trend x training x drug interaction ($F_{(1, 40)}$=0.001, p=0.98, 95% CI [−0.76, 0.74]).

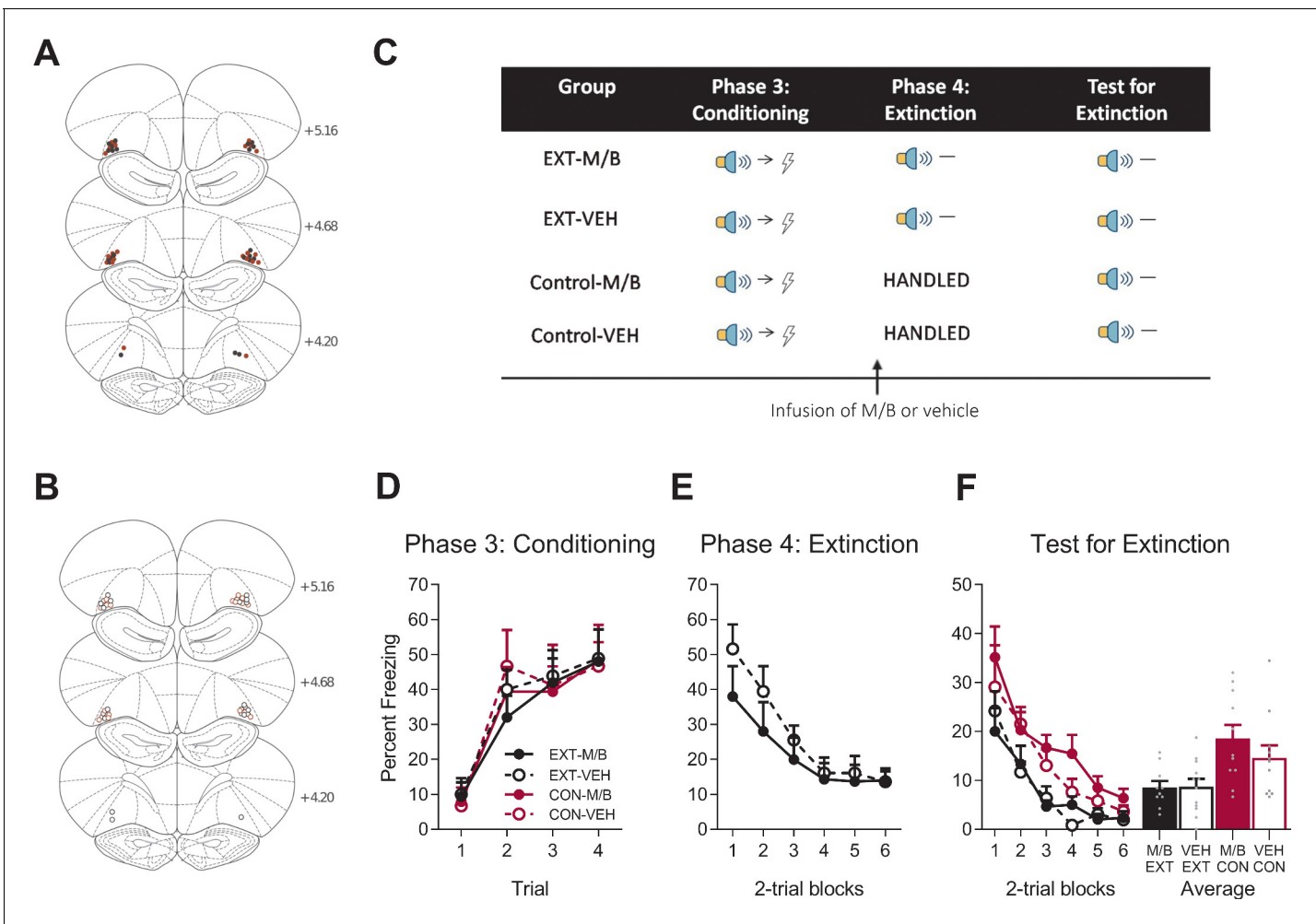

**Figure 4.** The lOFC is not necessary for extinction by omission. Using the same animals, the location of cannula placements reallocated for (A) drug- and (B) vehicle-infused rats in the lOFC extinction experiment as verified based on the atlas of *Paxinos and Watson, 1997*. The symbols represent the most ventral point of the cannula track for each rat and distances are indicated in millimetres from bregma. (C) Behavioural design for Extinction. Behavioural data are represented as mean + SEM percent levels of freezing during the cue period for (D) Conditioning, (E) Extinction, and (F) Test for Extinction of the target stimulus. Extinction-M/B (filled black), n = 10; extinction-vehicle (open black), n = 12; control-M/B (filled burgundy), n = 11; control-vehicle (open burgundy), n = 11.

The online version of this article includes the following source data for figure 4:

**Source data 1.** The lOFC is not necessary for extinction by omission.

Following this training, half of the cohort of rats received infusions of either M/B or vehicle into the lOFC prior to extinction training in which the cue was presented in the absence of the associated shock (*Figure 4C*). The remaining half of the cohort received identical infusions in the absence of extinction training. Infusion of M/B in the lOFC prior to extinction training had no effect on within-session performance (*Figure 4E*). Freezing to the cue across extinction did not differ between extinction-M/B and extinction-vehicle rats. A mixed ANOVA revealed no main effect of drug ($F_{(1, 20)}$=0.90, p=0.35, 95% CI [−0.85, 0.32]), a reduction in fear across extinction trials shown by a significant linear trend ($F_{(1, 20)}$=35.80, p<0.001, $\eta_p^2$ = 0.64, 95% CI [−1.55,−0.75]) and no linear trend x drug interaction ($F_{(1, 20)}$=2.33, p=0.14, 95% CI [−0.22, 1.38]), indicating that the rate of reduction in responding was similar for both extinction groups.

The following day, rats received presentations of the target cue in the absence of shock during Test. Overall, extinction training reduced fear on Test, but this extinction effect was not modulated by inactivation of the lOFC during extinction learning (*Figure 4F*). A mixed ANOVA revealed a main effect of training ($F_{(1, 40)}$=13.65, p=0.001, 95% CI [−0.88,−0.14], d = 1.12) but no main effect of drug ($F_{(1, 40)}$=0.79, p=0.38, 95% CI [−0.25, 0.50]), and no training x drug interaction ($F_{(1, 40)}$=0.90, p=0.35, 95% CI [−0.50, 0.24]). There was a significant linear trend indicating a reduction in conditioned freezing to the cue on Test across trials ($F_{(1, 40)}$=79.81, p<0.001, $\eta_p^2$ = 0.67, 95% CI [−1.53,−0.96]) but no linear trend x training x drug interaction ($F_{(1, 40)}$=0.19, p=0.67, 95% CI [−0.63, 0.88]), confirming a similar decrease in responding across trials for all groups. Taken together, these data provide clear evidence for a role of lOFC in fear reduction driven by overexpectation but not by extinction.

## Experiment 3: The role of the lOFC in initial and subsequent extinction

The results of Experiment 2 are important because they provide additional evidence to that presented in Experiment 1 that extinction and overexpectation are dissociable in the cortex (but see *Iordanova et al., 2016* for evidence of convergence in the central nucleus of the amygdala). However, our lack of an inactivation effect of the lOFC on extinction is somewhat at odds with previous studies showing that inactivation of this region disrupts extinction of fear (*Zimmermann et al., 2018*) and between-session extinction of reward associations (*Panayi and Killcross, 2014*). A critical difference between our finding and that of others is that the overexpectation preceded the extinction part of the experiment. Therefore, in order to obtain a pure assessment of the role of the lOFC in extinction, we implanted cannulae bilaterally in the lOFC (*Figure 5A and B*) and examined the effect of lOFC inactivation in extinction in the absence of any prior experience in Experiment 3. We also examined whether lOFC inactivation disrupts extinction in animals that have undergone extinction training previously with a different cue. This allowed us to assess the generality of the role of the lOFC in extinction.

Rats were trained to associate a cue (steady light or white-noise, counterbalanced) with footshock (*Figure 5C*). Conditioned fear was acquired to the cue (*Figure 5D*). A mixed ANOVA revealed no main effect of training ($F_{(1, 51)}$=2.72 p=0. 11, 95% CI [−0.81, 0.19]), no main effect of drug ($F_{(1, 51)}$=0.001, p=0.98, 95% CI [−0.50, 0.51]), and no training x drug interaction ($F_{(1, 51)}$=0.048, p=0.83, 95% CI [−0.54, 0.46]). Fear to the cue increased across trials revealed by a significant linear trend ($F_{(1, 51)}$=482.11, p<0.001, $\eta_p^2$=0.90, 95% CI [2.62, 3.14] ), and fear acquisition did not differ between the groups shown by the absence of a linear trend x training x drug interaction ($F_{(1, 51)}$=0.028, p=0.87, 95% CI [−0.66, 0.75]).

Following fear conditioning, rats were infused with M/B or vehicle into the lOFC and then trained in extinction, which consisted of presentations of the target cue alone in the absence of shock (*Figure 5C*). Rats in the control conditions received identical infusions but no extinction training (see Materials and methods). Infusions of M/B in the lOFC had no effect on the decline of freezing to the extinguished stimulus during extinction training (*Figure 5E*). A mixed ANOVA revealed no main effect of drug ($F_{(1, 30)}$=0.12, p=0.73, 95% CI [−0.42, 0.59]), a significant linear trend ($F_{(1, 30)}$=25.16, p<0.001, $\eta_p^2$ = 0.46, 95% CI [−0.92,−0.39]) and no linear trend x drug interaction ($F_{(1, 30)}$=0.42, p=0.52, 95% CI [−0.36, 0.70]).

To determine whether inactivation of the lOFC during extinction training had any effect on extinction learning, rats received test presentations of the target cue (tone or light, counterbalanced between rats) alone in the absence of shock (*Figure 5C*). Overall, extinction training reduced fear on Test (*Figure 5F*). A mixed ANOVA revealed a main effect of training ($F_{(1, 51)}$=56.68, p<0.001, 95% CI

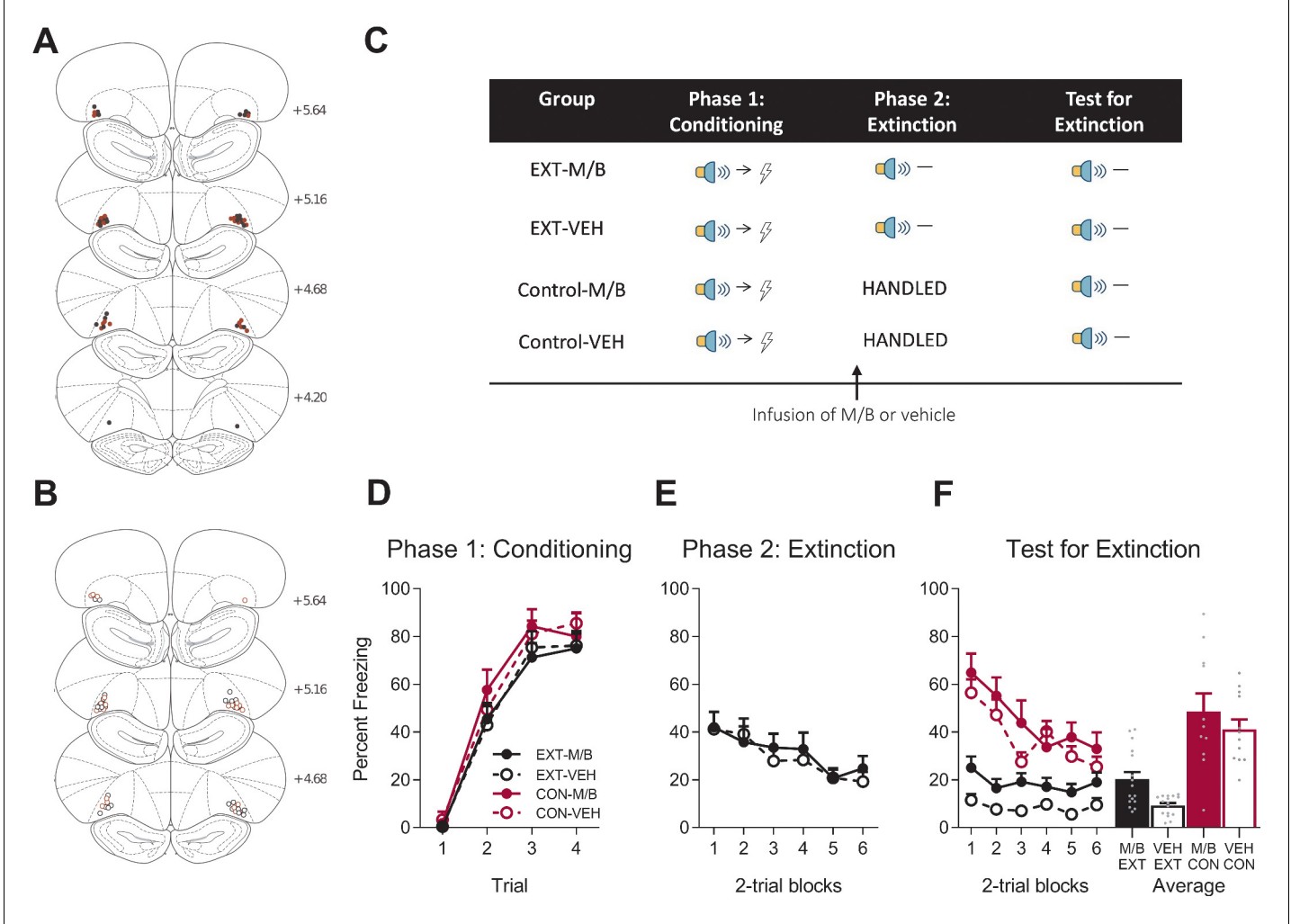

**Figure 5.** Inactivating the lOFC impairs but does not abolish extinction learning. Location of cannula placements for (A) drug- and (B) vehicle-infused rats in the lOFC extinction experiment as verified based on the atlas of *Paxinos and Watson, 1997*. The symbols represent the most ventral point of the cannula track for each rat and distances are indicated in millimetres from bregma. (C) Behavioural design for Extinction. Behavioural data are represented as mean + SEM percent levels of freezing during the cue period for (D) Conditioning, (E) Extinction, and (F) Test for Extinction of the target stimulus. Extinction-M/B (filled black), n = 16; extinction-vehicle (open black), n = 16; control-M/B (filled burgundy), n = 11; control-vehicle (open burgundy), n = 12.

The online version of this article includes the following source data for figure 5:

**Source data 1.** Inactivating the lOFC impairs but does not abolish extinction learning.

[−1.89,–0.96], d = 1.88), and a main effect of drug ($F_{(1, 51)}$=5.38, p=0.024, 95% CI [−0.03, 0.91], d = 0.43), but no training x drug interaction ($F_{(1, 51)}$=0.18, p=0.67, 95% CI [−0.39, 0.55]). Post hoc tests with Bonferroni adjustments revealed that a significant difference within the extinction condition: rats infused with M/B froze significantly more during presentations of the target stimulus at Test compared to vehicle rats ($F_{(1, 51)}$=4.51, p=0.039, 95% CI [−0.11, 1.15], d = 1.21); there was no effect of drug on the control condition ($F_{(1, 51)}$=1.54, p=0.22, 95% CI [−0.39, 1.11]). There was a significant linear trend across trials ($F_{(1, 51)}$=55.35, p<0.001, $\eta_p^2$ = 0.52, 95% CI [−0.89,–0.51]), confirming a decline in responding across Test, a linear trend x training interaction ($F_{(1, 51)}$=30.76, p<0.001, $\eta_p^2$ = 0.38, 95% CI [0.58, 1.50]) confirming that only the control groups showed a decline across Test, but there was no linear trend x training x drug interaction ($F_{(1, 51)}$=0.007, p=0.93, 95% CI [−0.54, 0.51]). These results are intriguing in that they reveal a disruption in extinction in animals trained with an inactivated lOFC. However, the data also show there is a clear effect of extinction in the inactivated animals relative to the controls, suggesting that lOFC inactivation during extinction resulted in a

mild and not catastrophic impairment in the retrieval of the extinction memory. This mild impairment is further put into context when considering the lack of change in the level of responding across test trials. The data suggest that both extinction groups are at performance floor. Therefore, the slightly higher level of responding in the extinction-drug animals compared to the extinction-vehicle animals is unlikely to be due to disruption in extinction learning or else we could see additional reduction in responding on Test.

Following initial extinction, the same rats took part in an additional extinction experiment (*Figure 6C*). The allocation of rats to groups was counterbalanced based on their prior experience in the extinction study (see Materials and methods) and their placements in this part of the experiment are depicted as per their new group assignment (*Figure 6A and B*). Acquisition of fear to the novel cue (flashing light or tone, counterbalanced) was equivalent between all groups (*Figure 6D*). A mixed ANOVA revealed no main effect of training ($F_{(1, 51)}$=3.25, p=0. 077, 95% CI [−0.81, 0.16]), no main effect of drug ($F_{(1, 51)}$=0.46, p=0.50, 95% CI [−0.61, 0.36]), and no training x drug interaction

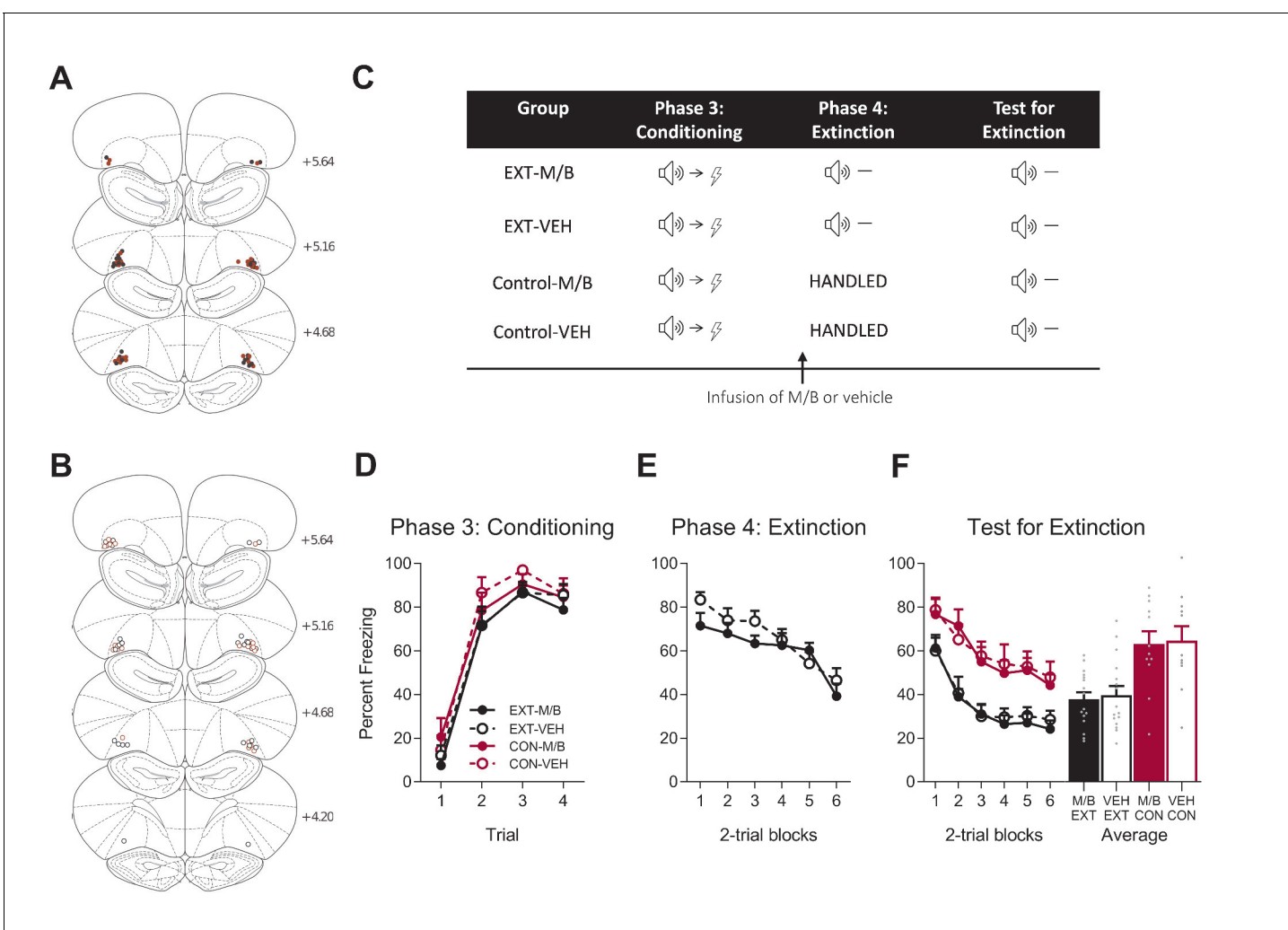

**Figure 6.** The lOFC is not required for subsequent extinction learning. Using the same animals, the location of cannula placements reallocated for (A) drug- and (B) vehicle-infused rats in the lOFC re-extinction experiment as verified based on the atlas of *Paxinos and Watson, 1997*. The symbols represent the most ventral point of the cannula track for each rat and distances are indicated in millimetres from bregma. (C) Behavioural design for Extinction. Behavioural data are represented as mean + SEM percent levels of freezing during the cue period for D) Conditioning, (E) Extinction, and F) Test for Re-Extinction of the target stimulus. Extinction-M/B (filled black), n = 16; extinction-vehicle (open black), n = 16; control-M/B (filled burgundy), n = 12; control-vehicle (open burgundy), n = 11.

The online version of this article includes the following source data for figure 6:

**Source data 1.** The lOFC is not required for subsequent extinction learning.

($F_{(1, 51)}$=0.001, p=0.98, 95% CI [−0.49, 0.48]). There was an increase in freezing across conditioning revealed by a significant linear trend ($F_{(1, 51)}$=297.22, p<0.001, $\eta_p^2$ = 0.85, 95% CI [2.24, 2.83]), and this increase did not differ between groups revealed by the absence of a linear trend x training x drug interaction ($F_{(1, 51)}$ = 0.090, p = 0.77, 95% CI [-0.70, 0.88]).

Following this training, half of the rats received infusions of either M/B or vehicle into the lOFC prior to extinction training in which the cue was presented in the absence of the associated shock (*Figure 6C*). The remaining rats received identical infusions in the absence of extinction training. Inactivation of the lOFC prior to extinction training had no effect on within-session performance and there were no differences in conditioned fear between extinction-M/B and extinction-vehicle rats (*Figure 6E*). A mixed ANOVA revealed no main effect of drug ($F_{(1, 30)}$=0.99, p=0.33, 95% CI [−0.65, 0.22]), a significant linear trend across trials ($F_{(1, 30)}$=57.16, p<0.001, $\eta_p^2$ = 0.66, 95% CI [−1.24,−0.71]) but no linear trend x drug interaction ($F_{(1, 30)}$=1.28, p=0.27, 95% CI [−0.24, 0.82]), indicating that the rate of reduction in responding was similar for both extinction groups.

The following day, all rats were tested for fear to the target cue. Overall, extinction training reduced fear on Test, but this extinction effect was not modulated by inactivation of the lOFC during extinction learning (*Figure 6F*). A mixed ANOVA revealed a main effect of training ($F_{(1, 51)}$=27.03, p<0.001, 95% CI [−1.47,−0.44], d = 1.40) but no main effect of drug ($F_{(1, 51)}$=0.16, p=0.74, 95% CI [−0.58, 0.45]), and no training x drug interaction ($F_{(1, 51)}$=0.001, p=0.97, 95% CI [−0.52, 0.51]). There was a significant linear trend indicating a reduction in freezing to the cue on Test across trials ($F_{(1, 51)}$=89.62, p<0.001, $\eta_p^2$ = 0.64, 95% CI [−1.19,−0.77]) but no linear trend x training x drug interaction ($F_{(1, 51)}$<0.001, p=0.99, 95% CI [−0.58, 0.58]), confirming a similar decrease in responding across trials for all groups. These findings confirm that inactivation of the lOFC with M/B prior to extinction training had no effect on subsequent extinction learning, replicating the findings of Experiment 2.

## Discussion

Our findings represent a fundamental contribution to the study of fear reduction because they define the conditions under which the IL and the lOFC are recruited. This was done by using extinction and overexpectation, two different experimental designs that attenuate learned fear. Our data confirm a well-established finding for the role of the IL in extinction of fear, but uncovered that the IL is not critical for all instances when learned fear is reduced. Specifically, inactivation of the IL during overexpectation learning left this effect intact. Inactivation of the lOFC, on the other hand, led to a catastrophic disruption of overexpectation but only a very mild, if any, actual impairment in extinction.

Our data replicate substantial prior work from the aversive domain showing that inactivation of the IL during extinction training disrupts extinction retrieval on Test (e.g., *Quirk et al., 2000*; *Milad and Quirk, 2002*; *Milad et al., 2004*; *Sierra-Mercado et al., 2006*; *Sierra-Mercado et al., 2011*; *Laurent and Westbrook, 2009*; *Do-Monte et al., 2015*; *Lingawi et al., 2017*). The effect of IL inactivation on performance during extinction training, however, has yielded variable results: Some studies show little to no effect on responding (e.g. *Do-Monte et al., 2015*), while others report a retardation (e.g., *Sierra-Mercado et al., 2011*) or facilitation (e.g., *Mendoza et al., 2015*; *Lay et al., 2019*). Our data are somewhat unclear in this regard. While we report a difference between the vehicle and inactivated extinction animals, this difference seems to be primarily driven by the controls, which show a reduced level of fear from conditioning to extinction training and compared to the vehicle extinction animals in the other experiments. While this difference is hard to interpret, our effect on Test is not, as revealed by the identical responding between the IL-inactivated animals relative to the non-extinction controls.

Our data on the role of the lOFC in extinction is particularly intriguing. We show an effect of lOFC inactivation only in initial extinction learning consistent with prior work (aversive: *Zimmermann et al., 2018*; appetitive: *Panayi and Killcross, 2014*) but not in subsequent extinction learning. However, our data provide an important control comparison that sheds fundamental light on the disruptive role of lOFC inactivation on initial extinction. lOFC inactivation during extinction did not prevent these animals from showing an extinction effect compared to the non-extinction controls and this difference was similar to that obtained from the vehicle animals. There was no additional reduction in conditioned responding during test in the extinction animals irrespective of drug

infusion, suggesting that they had reached floor performance. This result is key because it shows that disrupting lOFC function did not abolish the ability to attenuate learned fear under extinction conditions. This is further supported by a complete lack of effect of lOFC inactivation on subsequent extinction learning. Noteworthy is the fact that our subsequent extinction is not equivalent to re-extinction as we use novel cues and do not re-condition and re-extinguish experienced cues. While these data provide evidence that the lOFC has a very limited, if any, effect on extinction learning, they must be taken with caution because we observed a substantial drop in performance at the outset of extinction training (Phase 2) seen in *Figure 5* of Experiment 3 .

The dissociation of function reported here flies in the face of a parsimonious explanation for fear reduction in the IL and lOFC. Specifically, both methods are underpinned by the same negative prediction error mechanism (*Rescorla and Wagner, 1972*), which is generated when expectations surpass reality. While this common process can account for the behavioural effect of reduced fear on Test, it is unlikely to be what is regulated by the IL and lOFC, or else we would have seen similar disruptions in both behavioural designs. Instead, our findings suggest that the IL and the lOFC are recruited under different task-dependent conditions to downregulate fear.

To understand the function of these brain areas, it is important to highlight that the discrepancy between real and expected events is generated differently in the two designs. In extinction, the learning discrepancy results from omitting the delivery of an expected (aversive) event, which results in inhibition of fear responding, and likely underlies the development of inhibitory cue-response associations. This is unlikely to be the case in overexpectation because responding is maintained at high levels due to shock delivery. Thus, the pattern observed in the IL inactivation animals is consistent with inhibitory learning that likely involves cue-response associations. In contrast, the role of the lOFC in overexpectation and its limited role in extinction suggest that the lOFC is unlikely to be involved in this form of inhibitory learning. Rather, in overexpectation the learning discrepancy depends on the summation of two individually trained associations with the aversive event, therefore the lOFC likely modulates fear reduction through cue-outcome inhibitory learning. It is noteworthy, however, that the lOFC did have a small effect on initial extinction. While this leaves open the possibility that inhibitory cue-outcome associations might contribute to extinction, they are unlikely to drive the majority of response loss reported in our studies. Finally, the dissociation of function between the IL and lOFC is in line with data implicating IL activation during reduction of conditioned fear using a habituation method (*Furlong et al., 2016*), as well as research implicating the lOFC in cue-outcome associations (*Lichtenberg et al., 2017*; *Zhang and Li, 2018*, *Takahashi et al., 2009*, *Asok et al., 2013*).

Two further procedural differences between overexpectation and extinction warrant attention. The first relates to the number of trials necessary for learning in extinction and overexpectation. This difference is fundamental to the two designs and less easily subjected to analysis. Reducing the number of extinction trials can disrupt the effect and even create conditions for reconsolidation (e.g., *Eisenberg et al., 2003*; *Lee et al., 2006*), an altogether different phenomenon. On the other hand, increasing the number of compound presentations disrupts the overexpectation effect (*Garfield and McNally, 2009*). Therefore, these differences are inherent to the behavioural effects in fear. Nevertheless, it is worth noting that evidence from appetitive studies shows that when training is somewhat equivalent in overexpectation and extinction, the IL continues to have a dissociable role (*Lay et al., 2019*) as does the lOFC (*Takahashi et al., 2009*, *Burke et al., 2009*).

The second procedural difference relates to the mode of stimulus presentation. In extinction, animals learn about a single cue, whereas in overexpectation two different cues are presented in compound. To investigate the impact of this difference, it would be valuable to examine the contribution of the IL and lOFC to extinction learning using novel compounds comprising previously trained or pre-exposed cues. This line of research, however, carries many complexities. Stimuli presented in compound can interact in a variety of ways, one of which is summation (*Rescorla, 2000*; *Rescorla and Wagner, 1972*). As a result, nonreinforced compounds can facilitate (*Reberg, 1972*; *Rescorla, 2000*; *Leung and Westbrook, 2008*; *McConnell et al., 2013* or disrupt (*Rescorla, 2003*; *McConnell and Miller, 2010*) extinction learning, both of which likely depend on the lOFC. Relatedly, the lOFC may be required for keeping track of changes in associations across two or more stimuli, whether presented in compound or individually. These possibilities should be subjected to further investigation. In the present analysis, we focussed on examining the role of the

lOFC in the traditional extinction procedure that does not require cue interaction such as summation.

Our findings in fear mirror those in reward learning. Disruption of IL function interferes with appetitive extinction (natural rewards: *Rhodes and Killcross, 2004*; *Rhodes and Killcross, 2007a*; *Rhodes and Killcross, 2007b*; *Lay et al., 2019*; drug rewards: *Peters et al., 2008*; *LaLumiere et al., 2010*; for review of IL in fear and addiction see *Peters et al., 2008*) but not with appetitive overexpectation (*Lay et al., 2019*). Notably, a comparison across experiments suggests that the severity of the IL-dependent extinction deficit might be greater for the aversive (Experiment 1) relative to the appetitive case (*Lay et al., 2019*). Related to what may be learned in extinction, the IL has a role in appetitive and aversive response inhibition (*Capuzzo and Floresco, 2020*; *Martínez-Rivera et al., 2019*). In regard to the lOFC, electrophysiological and inactivation studies have linked it to appetitive overexpectation (*Takahashi et al., 2009*), but not to response inhibition when a conditioned cue is followed by reward omission (*Burke et al., 2009*, see also *Chudasama et al., 2007*). Our lOFC fear data are in line with these reports and join others (*Asok et al., 2013*; *Baker et al., 2018*; *Chang et al., 2018*; *Ray et al., 2018*; *Sarlitto et al., 2018*; *Trow et al., 2017*; *Zimmermann et al., 2018*) in providing evidence that the lOFC has a role beyond the appetitive domain.

To conclude, our data circumscribe the role of the IL in fear reduction and identify the lOFC as a novel avenue for research and clinical intervention in fear-related disorders (but see *Milad and Rauch, 2007*). In addition, they highlight the need to transcend the single-paradigm approach if a thorough understanding of the neural mechanisms supporting the reduction of conditioned responding is to be attained.

## Materials and methods

### Subjects

One hundred and sixty-seven (63 in Experiment 1, 48 in Experiment 2, 56 in Experiment 3) male Sprague Dawley rats weighing between 300–370 g were obtained from Harlan. Sample sizes were based on prior behavioural research in fear from our laboratory (*Mahmud et al., 2019*). Rats were pair-housed in a standard clear cage (44.5 cm x 25.8 cm x 21.7 cm) containing a mixture of beta chip and corncob bedding. The boxes were kept in an air-conditioned colony room maintained on a 12 hr light-dark cycle (lights off at 10:00 am). Food and water were available ad libitum prior to surgery and throughout the entire duration of the experiment. All experimental procedures were in accordance with the approval granted by the Canadian Council on Animal Care and the Concordia University Animal Care Committee.

### Surgery and drug infusion

Before behavioural training and testing, rats were implanted with bilateral guide cannulae in the IL or lOFC. Rats were anaesthetized with isoflurane gas and then mounted on a stereotaxic apparatus (David Kopf Instruments). They were then treated with a subcutaneous injection of 0.15 ml (50 mg/ml) solution of rimadyl (Pfizer, Kirkland, QC) immediately upon placement in the stereotaxic frame. Twenty-two-gauge single-guide cannulae (Plastics One) were implanted through holes drilled in both hemispheres of the skull above the IL or lOFC. The tips of the guide cannulae were aimed at the IL , the following coordinates were used: 2.9 mm anterior to bregma, 2.6 mm lateral to the midline at a 30° angle (bypassing the prelimbic cortex), and 4.2 mm ventral to bregma. For the lOFC using the following coordinates: 3.7 mm anterior to bregma, 2.7 lateral to the midline, and 4.3 ventral to bregma. The guide cannulae were secured to the skull with four jeweller's screws and dental cement. A dummy cannula was kept in each guide at all times except during microinjections. Rats were allowed six days to recover from surgery, during which time they were handled, weighed, and given an oral administration of 0.5 ml solution of cephalexin daily.

A cocktail of muscimol/baclofen or vehicle was infused bilaterally into the lOFC or IL by inserting a 28-gauge injector cannula into each guide cannula. The injector cannulas were connected to a 10 µL Hamilton syringe attached to an infusion pump (Harvard Apparatus). The injector cannula projected an additional 1 mm ventral to the tip of the guide cannula. A total volume of 0.3 µl was delivered to both sides at a rate of 0.1 µl/min, and drug delivery was monitored with the progression of

an air bubble in the infusion tube. The injector cannula remained in place for an additional 2 min after the infusion to allow for drug diffusion before its complete removal. Immediately after the infusion, the injector was replaced with the original dummy cannula. One day before infusions, all rats were familiarised with this procedure by removing the dummy cannula and inserting the injector cannula to minimise stress the following day.

## Drugs

A GABA$_A$ agonist, muscimol (M1523, Sigma-Aldrich), and GABA$_B$ agonist, baclofen (B5399, Sigma-Aldrich), were used to pharmacologically inactivate the IL and the lOFC. A muscimol-baclofen (M/B) cocktail was prepared by dissolving 5 mg of muscimol and 93.65 mg of baclofen in 438 ml of non-pyrogenic saline (0.9% w/v) to obtain a final stock concentration of 0.1 mM muscimol-1 mM baclofen. Saline was used as a vehicle solution.

## Histology

Subsequent to behavioural testing, rats received a lethal dose of sodium pentobarbital diluted 1:1 with 0.9% sodium chloride (120 mg/kg). The brains were removed and sectioned coronally at 40 μm through the IL or lOFC. Every second section was collected on a slide and stained with cresyl violet. The location of the cannulation tips was determined under a microscope using the boundaries defined by the atlas of *Paxinos and Watson, 1997*.

Rats with incorrect placements or infection were excluded from the statistical analyses. The numbers of rats excluded from each experiment based on incorrect placements or infection were fourteen in Experiment 1, three in Experiment 2, and one in Experiment 3. The fourteen rats that were excluded from Experiment 1 were used as anatomical controls and infusions of M/B outside of the IL cortex had no effect on the overexpectation effect ($F_{(1, 10)}$=0.63, p=0.45, 95% CI [−1.79, 1.01], mixed ANOVA) or extinction effect ($F_{(1, 7)}$=2.41, p=0.16, 95% CI [−2.35, 0.79], mixed ANOVA). One rat from Experiment 2 was identified as a significant outlier on Test for Overexpectation using the Grubb's outlier test (p<0.05) and thus excluded from all statistical analyses. The final *n*s for each group are presented in the figure legend for each experiment.

## Behavioural apparatus

### Experimental chambers

Behavioural procedures were conducted in eight operant-training chambers, each measuring 31.8 cm in height x 26.7 cm in length x 25.4 cm in width (Med Associates, St. Albans, VT, USA). The modular left and right walls were made of aluminium, and the back wall, front door, and ceiling were made of clear Perspex. Their floors consisted of stainless-steel rods, 4 mm in diameter, spaced 15 mm apart, center to center, with a tray below the floor. The grid floor was connected to a shock generator and delivered continuous scrambled foot-shock. Each chamber was enclosed in a ventilated sound attenuating cabinet. The back wall of each cabinet was equipped with a camera connected to a monitor located in another room of the laboratory where the behaviour of each rat was videotaped and observed by an experimenter. Illumination of each chamber was provided by a near-infrared light source (NIR-200) mounted on the back wall of each cabinet. Stimuli were presented through Med Associates software on a computer located outside the experimental room. The chambers had checkered or spotted wallpaper on the door and each wall with the exception of the back wall to allow for video viewing. Instead, the back wall of the holding cabinet was covered in either checkered or spotted wallpaper. The chambers (walls, ceiling, door, grid floor, and tray) were cleaned with 4% almond-scented solution (PC Black Label) after the removal of each rat. The chamber wallpaper was counterbalanced across groups.

### Stimuli

Visual and auditory cues were used in the experiments. The visual cues consisted of a 4 Hz flashing light located on the left-hand side of the right wall and a steady light located on the right-hand side of the right wall. The auditory cues were a 70 dB tone and a 72 dB white-noise (measured inside the chamber) delivered through a loud speaker located outside the behavioural chamber. The background noise in the chamber was 48–50 dB. The background noise and experimental auditory cues were measured using a digital sound level meter (Tenma, 72–942). All cues were 30 s in duration

and were fully counterbalanced. During Phases 1, 2, and 3, the cues terminated with the onset of a 1 s duration foot-shock at 0.5 mA intensity that was delivered to the floor of each chamber. The cues were controlled via a Med-Associates program.

## Behavioural procedures

Each study was done in two replications. Allocation to groups was based on Conditioning (Phases 1 and 3) to ensure that all groups entered the critical part of training (Overexpectation and Extinction for Experiments 1 and 2, Initial Extinction and Subsequent Extinction for Experiment 3) with the same level of responding. The behavioural sequence for Experiments 1 and 2 is depicted in *Figure 7A* and Experiment three in *Figure 7B*.

## Experiments 1 and 2

### Phase 1 Conditioning

On days 1 to 6, rats were placed into the conditioning context for 20 min sessions. Following a 9 min 30 s adaption period, all rats received one paired presentation of the cue terminating with a shock (tone on days 1, 3 and 5; flashing light on days 2, 4, and 6). On days 4 to 6, three hours following conditioning, rats were placed back in the conditioning context for 20 min and no cues were presented. These context extinction sessions were carried out in order to reduce any fear to the background cues and thus allow for a clearer assessment of the acquisition of freezing to the cues.

### Phase 2 Overexpectation Training

Following Phase 1 Conditioning, rats were assigned to either an overexpectation or control condition based on their responding during Conditioning. Thirty minutes prior to the start of behavioural training in Phase 2, rats received pre-training infusion of M/B or vehicle into the IL (Experiment 1) or lOFC (Experiment 2). Following infusions, rats were further assigned to either an overexpectation or control condition, which yielded four sub conditions: overexpectation-M/B, overexpectation-vehicle, control-M/B, and control-vehicle.

On days 7, rats in the overexpectation condition were placed in the conditioning context and received two compound presentations of the tone and flashing light terminating with the onset of a shock (0.5 mA, 1 s). The first trial began 4 min 30 s after being placed in the context and intertrial intervals (ITI) were 4 min 30 s. Rats remained in the chamber for 5 min following the final compound presentation. Rats in the control condition were handled for 30 s in their home-cage.

### Test for Overexpectation

On day 8, rats were tested for responding to either the visual (flashing light) or the auditory (tone) cue (counterbalanced). All rats were placed in the conditioning context, and after a 5 min adaptation period, the cue was presented. The test session consisted of eight stimulus alone presentations with an ITI of 1 min.

### Phase 3 Conditioning

On day 9, rats were placed in the conditioning context, and after a 5 min adaption period, received four, 30 s paired presentations of a novel cue (steady light or white-noise, counterbalanced across rats) and shock (0.5 mA, 1 s) with an ITI of 3 min. Rats remained in the chamber for 2 min following the final stimulus presentation.

### Phase 4 Extinction

Following Phase 3 Conditioning, rats were reassigned to either an extinction or control condition based on their responding during Conditioning. Rats were then further assigned to either the drug or vehicle condition such that half the rats that had previously received an infusion of the drug now received vehicle, whereas the other half received drug, and similarly for rats that were initially allocated to the vehicle condition. This yielded four sub-conditions: extinction-M/B, extinction-vehicle, control-M/B, and control-vehicle. Infusions of M/B or vehicle into IL (Experiment 1) or the lOFC (Experiment 2) occurred 30 min prior to the start of the extinction session.

On day 10, rats in the extinction condition were placed in the conditioning context, and after a 5 min adaption period, the cue was presented. The extinction session consisted of twelve, 30 s

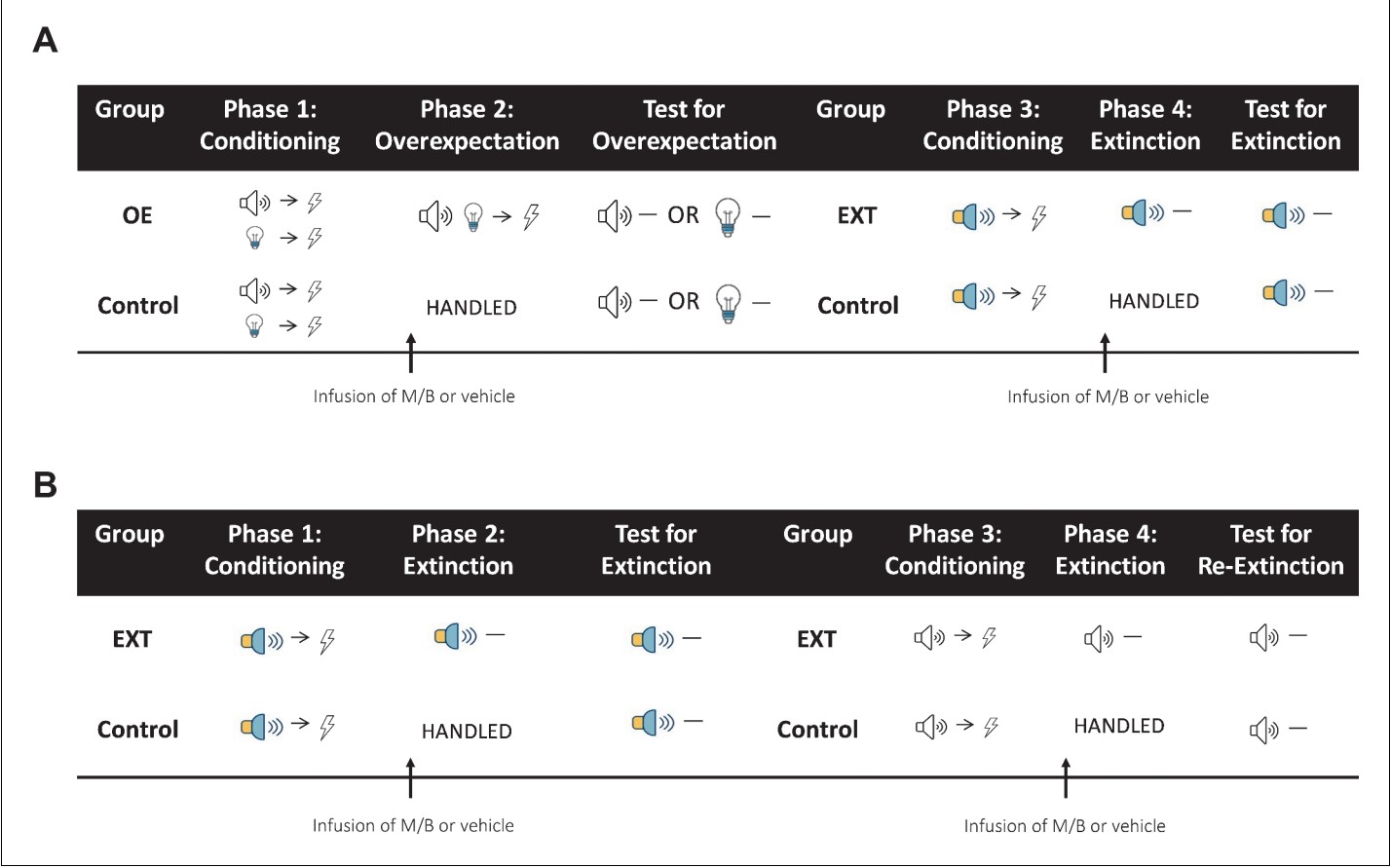

**Figure 7.** Behavioural sequence for Experiments 1–3.  (**A**) In Experiments 1 and 2, rats are trained to associate two individual cues (tone and light) with a shock during Phase 1 and their performance during this phase was used to determine group allocation such that fear acquisition was similar between the groups. Immediately prior to overexpectation training (Phase 2), the IL (Experiment 1) or lOFC (Experiment 2) was pharmacologically silenced with muscimol and baclofen (0.1 mM muscimol-1 mM baclofen, M/B). Rats in the overexpectation group (OE) received compound presentations of the two cues followed by the delivery of a shock. Rats in the control group (Control) were handled. All rats were then tested for conditioned responding (freezing) the following day to either the tone or the light (counterbalanced). Following the overexpectation experiment, the same rats received conditioning to a novel stimulus (white-noise or steady light, counterbalanced) paired with shock during Phase 3. Rats were reassigned to one of the four groups (counterbalanced for training and drug history) based on their responding during Conditioning. Prior to extinction in Phase 4, rats received an infusion of M/B or vehicle into the IL or lOFC. Rats in the extinction condition (EXT) received non-reinforced presentations of the target cue. Rats in the control condition (Control) were handled. All rats were then tested for conditioned responding to the target cue (counterbalanced) the following day. (**B**) In Experiment 3, rats were trained to associate a cue (steady light or white-noise, counterbalanced between rats) with shock during Phase 1. Prior to extinction training (Phase 2), rats received an infusion of M/B or vehicle into the lOFC. Rats in the extinction condition (EXT) received non-reinforced presentations of the fear-conditioned cue. Rats in the control condition (Control) were handled. All rats were then tested for conditioned responding to the target cue (counterbalanced) the following day. Following initial extinction, the same rats took part in a subsequent extinction experiment. Rats received conditioning to a novel stimulus (tone or flashing light, counterbalanced) paired with shock during Phase 3. Rats were reassigned to groups based on their responding during Conditioning. Immediately prior to extinction training in Phase 4, the lOFC was pharmacologically silenced in the same manner as described above. Again, rats in the extinction condition (EXT) received non-reinforced presentations of the target stimulus while rats in the control condition (Control) were handled. All rats were then tested for conditioned responding to the target cue the following day.

stimulus alone presentations with an ITI of 3 min (see *Do-Monte et al., 2015*). Rats remained in the chamber for 2 min following the final stimulus presentation. Rats in the control condition were handled for 30 s in their home-cage.

## Test for Extinction
On day 11, all rats were tested drug-free for responding to the extinguished stimulus in a manner identical to that of the extinction session.

## Experiment 3

### Phase 1 Conditioning

On day 1, rats were placed in the conditioning context, and after a 5 min adaption period, received four, 30 s paired presentations of a steady light or a white-noise (counterbalanced across rats) and shock (0.5 mA, 1 s). The ITI between paired CS-shock presentations was 3 min. Rats remained in the chamber for 2 min following the final stimulus presentation. Three hours following Conditioning, rats were placed back in the conditioning context for 20 min a no cues were presented. These context extinction sessions were carried out in order to reduce any fear to the background cues and thus allow for a clearer assessment of the acquisition of freezing to the cues.

### Phase 2 Extinction

Following Phase 1 Conditioning, rats were assigned to either an extinction or control condition based on their responding during Conditioning. Rats were then further assigned to either the drug or vehicle condition. This yielded four sub-conditions: extinction-M/B, extinction-vehicle, control-M/B, and control-vehicle. Infusions of M/B or vehicle into the lateral OFC occurred 30 min prior to the start of the extinction session.

On day 2, rats in the extinction condition were placed in the conditioning context, and after a 5 min adaption period, the cue was presented. The extinction session consisted of twelve, 30 s stimulus alone presentations with an ITI of 3 min. Rats remained in the chamber for 2 min following the final stimulus presentation. Rats in the control condition were handled for 30 s in their home-cage.

### Test for Extinction

On day 3, all rats were tested drug-free for responding to the extinguished stimulus in a manner identical to that of the extinction session.

### Phase 3 Conditioning

On day 4, rats were placed in the conditioning context, and after a 5 min adaption period, received four, 30 s paired presentations of a novel cue (flash or tone, counterbalanced across all rats) and foot-shock (0.5 mA, 1 s). This session was identical to that described for Phase 1. Three hours following Conditioning, rats were placed back in the conditioning context for 20 min where no cues were presented.

### Phase 4 Extinction

Following Phase 3 Conditioning, rats were reassigned to either the drug or vehicle condition such that half the rats that had previously received an infusion of the drug now received vehicle, whereas the other half received drug, and similarly for rats that were initially allocated to the vehicle condition. This yielded four sub-conditions: extinction-M/B, extinction-vehicle, control-M/B, and control-vehicle. Infusions of M/B or vehicle into the lateral OFC occurred 30 min prior to the start of the extinction session.

On day 2, rats in the extinction condition were placed in the conditioning context, and after a 5 min adaption period, the cue was presented. The number of trials, cue and ITI duration were identical to those described for Phase 2. Rats in the control condition were handled in the same manner as described previously.

### Test for Extinction

On day 6, all rats were tested drug-free for responding to the extinguished stimulus in a manner identical to that of the Test for Extinction session.

## Data analysis

Freezing was used to assess conditioned fear. It was defined as the absence of all movements except those related to breathing (*Blanchard and Blanchard, 1969*; *Fanselow, 1980*) and exhibited a hunched posture. Each rat was observed every 2 s and scored as either freezing or not freezing by two observers, one of whom was blind to group assignment. The correlation between the scores were high (Pearson r > 0.9). A percentage score was calculated for the proportion of the total

observation each rat spent freezing during the total duration of each stimulus presentation. The levels of freezing do not include freezing during the ITI. Data were analysed in SPSS 25.0 (IBM, New York, USA) using repeated measures analysis of variance (ANOVA). Significance was set at α = 0.05. Where appropriate, adjustments with Bonferroni were made for multiple comparisons. Effects of trial, where reported, were measured with contrasts testing for the presence of a linear trend. Standardized confidence intervals (CIs; 95% for the mean difference) and measures of effect size ($\eta_p^2$ for ANOVA and Cohen's d for contrasts; see *Cohen, 1988*) are reported for each significant comparison. Data for each experiment and phase were reported using all trials. Significant outliers were detected using the Grubbs outlier test (https://www.graphpad.com/quickcalcs/Grubbs1.cfm).

## Acknowledgements

This work was supported by FRQNT Nouveaux Chercheurs grant (2017-NC-198182, to MDI); a NARSAD Young Investigator grant (to MDI); a CIHR Project Grant (to MDI); the Canada Research Chairs program (to MDI); a FRQS post-doctoral fellowship (to BPPL); and a NSERC Undergraduate Student Research Award (to NB). The authors report no conflict of interest. All correspondence to be addressed to Dr. Mihaela D Iordanova (mihaela.iordanova@concordia.ca).

## Additional information

### Competing interests

Mihaela D Iordanova: Reviewing editor, *eLife*. The other authors declare that no competing interests exist.

### Funding

| Funder | Grant reference number | Author |
|---|---|---|
| Fonds de Recherche du Québec - Nature et Technologies | 2017-NC-198182 | Mihaela D Iordanova |
| Canadian Institutes of Health Research | Project Grant | Mihaela D Iordanova |
| Brain and Behavior Research Foundation | NARSAD grant | Mihaela D Iordanova |
| Canada Research Chairs | | Mihaela D Iordanova |
| Fonds de Recherche du Québec - Santé | | Belinda PP Lay |
| Natural Sciences and Engineering Research Council of Canada | | Nathan Boulianne |

The funders had no role in study design, data collection and interpretation, or the decision to submit the work for publication.

### Author contributions

Belinda PP Lay, Conceptualization, Data curation, Formal analysis, Writing - original draft; Audrey A Pitaru, Nathan Boulianne, Data curation, Investigation; Guillem R Esber, Writing - review and editing; Mihaela D Iordanova, Conceptualization, Formal analysis, Supervision, Funding acquisition, Methodology, Project administration, Writing - review and editing

### Author ORCIDs

Mihaela D Iordanova https://orcid.org/0000-0001-6232-448X

### Ethics

Animal experimentation: All experimental procedures were in accordance with the approval granted by the Canadian Council on Animal Care and the Concordia University Animal Care Committee.

Decision letter and Author response

Decision letter https://doi.org/10.7554/eLife.55294.sa1

Author response https://doi.org/10.7554/eLife.55294.sa2

## Additional files

### Data availability

All data generated or analysed during this study are included in the manuscript and supporting files. Source data files have been provided for all figures.

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
