## [Decision Letter]

**Acceptance summary:**

All reviewers recognized that a substantial effort has been made to address earlier comments. In particular, the inclusion of the new experiment 3 and the considerably revised Discussion have been well appreciated.

Regarding the new experiment, the shift to extinction appears to have brought about a massive reduction in performance, which casts a very small cloud over the claim that the OFC "impairs but does not abolish extinction learning" but overall, all reviewers recognized that the present work is a nice and valuable contribution to the literature documenting the cortical bases of fear reduction and could thus be accepted in its present form.

**Decision letter after peer review:**

Thank you for submitting your article "Different forms of fear extinction are supported by distinct cortical substrates" for consideration by *eLife*. Your article has been reviewed by three peer reviewers, including Mathieu Wolff as the Reviewing Editor and Reviewer #1, and the evaluation has been overseen by Kate Wassum as the Senior Editor. The following individual involved in review of your submission has agreed to reveal their identity: Gregory Quirk (Reviewer #3).

The reviewers have discussed the reviews with one another and the Reviewing Editor has drafted this decision to help you prepare a revised submission.

Summary:

The paper by Lay et al. uses rats to examine the roles of the infralimbic cortex (IL) and lateral orbitofrontal cortex (lOFC) in extinction and overexpectation of Pavlovian conditioned fear, relying on reversible inactivation of these cortical areas in rats. Using two distinct paradigm to assess fear extinction, the authors provide convincing support for a double dissociation at the cortical level: the infralimbic cortex appears to be necessary for extinction by omission (i.e. the most classic way to assess extinction), but not by overexpectation, while the lateral orbitofrontal cortex seems to support extinction by overexpectation but not omission. These results are important for understanding the dissociability of these two forms of fear reduction as well as the differential contribution of cortical subregions to fear reduction. This double dissociation was recognized as original and of interest for the field by the three reviewers. Some issues related to the interpretation of the data however require clarification before the work can be considered for publication. We produce below a list of essential revisions that the authors must address, together with a number of smaller issues that the authors are encouraged to consider.

Essential revisions:

1) Research Article

Since addressing some of the points will require to expand the Discussion and/or to include additional analyses, we suggest the authors to submit their revision as a Research Article, not a brief communication. This way, all necessary information can be conveniently incorporated into the main text. Statistical analyses in particular could therefore be presented in full where relevant (as data are being described, instead of at the end of the document as is presently the case).

2) Tasks order

The order of the task is a likely confound to interpret the data, especially for the extinction experiment. The within-experiment nature of the assessments conducted here indeed means that there was no pure assessment of the role of the OFC in initial extinction. What has been done could be interpreted as some assessment of the role of the OFC in re-extinction (especially as the stimuli used in the second part of each experiment were in the same modality as those used in the first part), which has been shown to differ in important respects from initial extinction. While some additional experiments could possibly address this issue, it may not be mandatory if the authors can maybe provide some additional analyses of freezing to strengthen the interpretations. This could possibly include freezing during the ITI, or prior to the first tone, or anything that could clarify this issue. It is clear that a more thorough Discussion around these points is expected in the revised paper.

3) Interpretation of the data

In line with the above, the extinction and over-expectation sessions differed in numerous respects other than the presence vs. absence of the shock: e.g., (1) number of stimuli presented, (2) number of presentations of those stimuli, (3) presentations of stimuli in compound, etc. These differences could well contribute to the differential effects of the IL and OFC manipulations on the two protocols that have been examined, and it would be worth knowing whether this is the case or not, and if so, exactly how/why. So again, additional analyses may help to address these issues which should also be properly discussed. When doing the latter, please consider that the language/descriptions used at several places in the paper could be revised to improve the clarity of its central message, and to avoid the potential over-stating of findings: (i) the second-half of the final paragraph is a bit distracting. (ii) The authors seem to imply that overexpectation is a form of extinction, but it could also be considered learned fear reduction as extinction may really imply omission of the US. For this reason, the word "extinction" in the title could be "reduction". The first paragraph of the main text also has this problem.

4) Overexpectation data

Comparison of Figures 1F and 3F seems to show a problem with the overexpectation. Figure 1F shows a significant reduction in freezing by overexpectation in SAL groups, and this is supported by the stats in the figure legend (Learning from overexpectation was successful (F(1, 45) = 19.21, p < 0.001, 95% CI [-1.48, -0.41], d = 1.30, mixed ANOVA). However, in Figure 3F, there seems to be a much smaller difference in the SAL groups, i.e. no clear effect of overexpectation training. The figure legend in 3F has no analogous statistical statement as in 1F. Only that there was a significant increase in the MUS group. The authors need to clarify the analyses here (more animals are needed?).

---

## [Author Response]

Essential revisions:1) Research ArticleSince addressing some of the points will require to expand the Discussion and/or to include additional analyses, we suggest the authors to submit their revision as a Research Article, not a brief communication. This way, all necessary information can be conveniently incorporated into the main text. Statistical analyses in particular could therefore be presented in full where relevant (as data are being described, instead of at the end of the document as is presently the case).

The manuscript has been revised in accordance with the guidelines for a Research Article such that the statistical analyses for each experiment are presented in full where relevant in the Results section. In addition, we have included a table that includes all analyses that were done.

2) Tasks orderThe order of the task is a likely confound to interpret the data, especially for the extinction experiment. The within-experiment nature of the assessments conducted here indeed means that there was no pure assessment of the role of the OFC in initial extinction. What has been done could be interpreted as some assessment of the role of the OFC in re-extinction (especially as the stimuli used in the second part of each experiment were in the same modality as those used in the first part), which has been shown to differ in important respects from initial extinction. While some additional experiments could possibly address this issue, it may not be mandatory if the authors can maybe provide some additional analyses of freezing to strengthen the interpretations. This could possibly include freezing during the ITI, or prior to the first tone, or anything that could clarify this issue. It is clear that a more thorough Discussion around these points is expected in the revised paper.

We completely agree with the reviewers that the task order presented a potential confound for the lOFC extinction experiment in particular (not so for the IL study because these extinction data replicated those reported by others) and that a pure assessment of the role of the lOFC in initial extinction was necessary. As we deemed this to be fundamental, we decided to run the experiment. This additional study (Experiment 3) assessed the role of the OFC in initial extinction and during subsequent extinction (as in Experiments 1 and 2). Our results show that lOFC inactivation had a disruptive effect on initial extinction similar to that reported by Zimmerman et al., 2018. However, our inclusion of non-extinction controls showed that this disruption was not indicative of a lack of extinction learning. Responding in the lOFC-inactivated extinction animals was much lower compared to the non-extinction controls, we did not get a significant interaction, and there was no additional decline in responding on test suggesting that no further extinction took place across the non-reinforced trials on test. This very modest effect of lOFC inactivation in extinction is clearly very distinct to the catastrophic effect of IL inactivation on extinction (as well as the catastrophic effect of lOFC inactivation on overexpectation). We also replicated our lack of lOFC inactivation on subsequent extinction. We address this in the Discussion by discussing what the likely role of the lOFC may be and what types of associations are likely to drive the reduction in conditioned fear in extinction (and overexpectation).

3) Interpretation of the dataIn line with the above, the extinction and over-expectation sessions differed in numerous respects other than the presence vs. absence of the shock: e.g., (1) number of stimuli presented, (2) number of presentations of those stimuli, (3) presentations of stimuli in compound, etc. These differences could well contribute to the differential effects of the IL and OFC manipulations on the two protocols that have been examined, and it would be worth knowing whether this is the case or not, and if so, exactly how/why. So again, additional analyses may help to address these issues which should also be properly discussed. When doing the latter, please consider that the language/descriptions used at several places in the paper could be revised to improve the clarity of its central message, and to avoid the potential over-stating of findings: (i) the second-half of the final paragraph is a bit distracting. (ii) The authors seem to imply that overexpectation is a form of extinction, but it could also be considered learned fear reduction as extinction may really imply omission of the US. For this reason, the word "extinction" in the title could be "reduction". The first paragraph of the main text also has this problem.

We agree with the reviewers that there are noteworthy procedural differences other than the presence and absence of shock that may account for the differential cortical effects. We address those in two paragraphs in the Discussion. While the number of trials are inherent to the tasks in fear, there is no evidence from the appetitive literature (where more trials are often necessary for learning) that trial number determines IL or OFC involvement. We also explain in the Discussion that decreasing the extinction trials would likely lead to reconsolidation, which would be valuable to explore but is beyond the scope of the current investigation. Increasing the overexpectation trials has been shown to disrupt the overexpectation effect (Garfield and McNally, 2009). The role of compound presentation is a particularly interesting question. We actually think that the lOFC should have a role in regulating extinction when compounds are used because of the role of summation in associative strength between cues and the competition for extinction learning between cues. To elaborate on this point, presenting two previously trained cues together in compound and not reinforcing the compound leads to potentiated extinction, this will likely be regulated by the lOFC because we think that, like in overexpectation, the lOFC will be necessary for summation and thus augmentation of the error term. Presenting a compound that consists of a reinforced and a non-reinforced cue can result in protection from extinction, which may also depend on the lOFC because of the competition for development in inhibitory associative strength, and if the non-reinforced cue is an inhibitor (true protection from extinction) then again summation (of excitatory and inhibitory associative strengths) is likely to be modulated by the lOFC. This is related to the number to stimuli presented as the lOFC may be required to keep track of the changes in associations across stimuli. We are 100% committed to conducting these studies but we are not able to do this as part of the present investigation. Here we wanted to focus on the more traditional procedures used in the field. To further understand the role of the lOFC in fear reduction we will undertake these hybrid studies of extinction and overexpectation as soon as the pandemic conditions ease and our institution allows for animal orders.

As a result of the reviewers’ comments and the potential stimulus compounding effects in extinction that could be lOFC-dependent, we have modified the narrative of the manuscript to focus on fear reduction and not on different types of extinction and the presence vs. absence of shock. This has been done throughout the manuscript and in the title. We have also emphasized the likely involvement of the lOFC in summation conditions. We thank the reviewers for this and we agree that this change was an excellent idea, while still preserving the main point of the manuscript that downregulation in learned fear can be obtained in distinct ways and this is dissociable in the IL and lOFC.

4) Overexpectation dataComparison of Figures 1F and 3F seems to show a problem with the overexpectation. Figure 1F shows a significant reduction in freezing by overexpectation in SAL groups, and this is supported by the stats in the figure legend (Learning from overexpectation was successful (F(1, 45) = 19.21, p < 0.001, 95% CI [-1.48, -0.41], d = 1.30, mixed ANOVA). However, in Figure 3F, there seems to be a much smaller difference in the SAL groups, i.e. no clear effect of overexpectation training. The figure legend in 3F has no analogous statistical statement as in 1F. Only that there was a significant increase in the MUS group. The authors need to clarify the analyses here (more animals are needed?).

We apologize for the omission in Figure 3F. We have corrected this and reported the relevant statistics in the text: ‘Rats infused with vehicle prior to overexpectation training froze significantly less during presentations of the target stimulus at Test compared to vehicle-treated rats in the control condition (*F*_(1, 40)_ = 5.93, *p* = 0.019, 95% CI [-1.63, 0.06], d = 1.26).’ (Figure 3, subsection “Experiment 2: The lateral OFC regulates learning from overexpectation but not extinction”).